# Design and Validation of a Gene-Targeted, Next-Generation Sequencing Panel for Routine Diagnosis in Gliomas

**DOI:** 10.3390/cancers11060773

**Published:** 2019-06-04

**Authors:** Nicky D’Haene, Bárbara Meléndez, Oriane Blanchard, Nancy De Nève, Laetitia Lebrun, Claude Van Campenhout, Isabelle Salmon

**Affiliations:** Department of Pathology, Erasme Hospital, Université Libre de Bruxelles, 1070 Brussels, Belgium; oriane.blanchard@erasme.ulb.ac.be (O.B.); nancy.de.neve@erasme.ulb.ac.be (N.D.N.); laetitia.lebrun@erasme.ulb.ac.be (L.L.); claude.van.campenhout@erasme.ulb.ac.be (C.V.C.); isabelle.salmon@erasme.ulb.ac.be (I.S.)

**Keywords:** next-generation sequencing, glioma, 1p/19q codeletion, molecular pathology

## Abstract

The updated 2016 World Health Organization (WHO) classification system for gliomas integrates molecular alterations and histology to provide a greater diagnostic and prognostic utility than the previous, histology-based classification. The increasing number of markers that are tested in a correct diagnostic procedure makes gene-targeted, next-generation sequencing (NGS) a powerful tool in routine pathology practice. We designed a 14-gene NGS panel specifically aimed at the diagnosis of glioma, which allows simultaneous detection of mutations and copy number variations, including the 1p/19q-codeletion and Epidermal Growth Factor Receptor (*EGFR*) amplification. To validate this panel, we used reference mutated DNAs, nontumor and non-glioma samples, and 52 glioma samples that were previously characterized. The panel was then prospectively applied to 91 brain lesions. A specificity of 100% and sensitivity of 99.4% was achieved for mutation detection. Orthogonal methods, such as in situ hybridization and immunohistochemical techniques, were used for validation, which showed high concordance. The molecular alterations that were identified allowed diagnosis according to the updated WHO criteria, and helped in the differential diagnosis of difficult cases. This NGS panel is an accurate and sensitive method, which could replace multiple tests for the same sample. Moreover, it is a rapid and cost-effective approach that can be easily implemented in the routine diagnosis of gliomas.

## 1. Introduction

Studies performed by the scientific community over the past two decades have substantially clarified the genetic basis of tumorigenesis in both common and some rarer central nervous system (CNS) tumor entities [1,2,3,4]. This improved understanding of brain tumors contributes to their classification and identifies molecular biomarkers that are useful in the diagnosis, prognosis, and therapeutic approach for treating patients with CNS tumors.

The updated 2016 World Health Organization (WHO) classification system of CNS tumors integrates well-established molecular parameters with histology to define a number of tumor entities [5]. In gliomas, depending on the tumor histopathology and the patient’s clinical features, this integrated diagnosis may require assessing the tumor for *IDH1* and *IDH2* (*IDH*), *H3F3A*, and *BRAF* mutations, as well as the 1p/19q codeletion [5]. Other key molecular biomarkers may have diagnostic utility, such as the *TERT*, *ATRX*, *TP53*, and *EGFR* genes [6,7,8,9,10], although those have been less extensively used in routine clinical practice. The difficulties of the technical implementation, the controversial clinical utility, and the relatively recent discovery of some of these markers could contribute to why they are still not being included in the updated WHO classification.

The codeletion of the chromosomal regions 1p and 19q characterize a particular subtype of glioma: the *IDH*-mutant, 1p/19q codeleted oligodendroglioma, as defined by the 2016 WHO classification. Tumors with these genetic alterations are associated with enhanced chemo- and radiosensitivity and a favorable prognosis [11,12]. The 1p/19q codeletion is tested for in most pathology laboratories through the use of fluorescence in situ hybridization (FISH), and the advantages and disadvantages of this technique compared to others have been previously discussed [13,14].

Next-generation sequencing (NGS) is increasingly being used in routine clinical practice, including for the diagnosis of rare entities like gliomas [15,16,17,18], because it can replace multiple single-gene genomic testing technologies while requiring only one test. Gene-targeted NGS offers a cost-effective approach to simultaneously detect multiple genetic alterations with a minimal amount of DNA and with high sensitivity, which makes this method highly attractive for use in gliomas. Panels that are specifically designed for gliomas are needed for the routine diagnosis of these tumors. Currently, many institutions use very large or non-glioma-specific panels that broadly serve all types of cancers.

In this report, we describe the validation of a custom glioma NGS-targeted panel that was developed at our institution to simultaneously detect both relevant glioma-related gene mutations and copy number variations (CNVs). We report here our one-year experience of clinical practice using this targeted glioma NGS panel to guide diagnosis.

## 2. Results

### 2.1. Sequencing Performance, Quality, and Turnaround Time

Sequencing performance was assessed by the number of sequencing reads across the targeted regions. A total of 146 NGS assays of the 143 samples were performed (three samples were repeated to confirm the findings). According to our quality criteria of the total series, 135 (92%), 7 (5%), and 4 (3%) samples showed optimal, suboptimal, and low quality, respectively. Low-quality samples were excluded from further analyses. The total mean coverage depth was 2007× (range 605–5132×), with similar coverage in the validation and prospective series (1894× and 2070×, respectively). Mean uniformity was 84% (range 18–93%). To note, samples showing *EGFR* amplification had the lowest uniformity (Appendix A).

For prospective samples, the median turnaround time (TAT) between reception of the sample in the laboratory and report release was 14 calendar days. The report was released within adequate TAT (18 calendar days) in 82% of the samples, with an average of 10 calendar days for these samples. After implementation of the panel into the routine workflow, the TAT of the other panels used in our laboratory was not influenced (data not shown).

### 2.2. Detection of Mutations with the Glioma Next-Generation Sequencing Panel

To assess the specificity of the NGS glioma panel, we first performed the analyses on 12 non-tumor samples. No mutations were detected in these samples, resulting in a specificity of 100%. To evaluate the sensitivity of the panel (i.e., to evaluate its capacity to detect gene mutations), we initially analyzed two commercially available reference standards: HDx Reference Standard (HDx, Horizon Discovery, Cambridge, UK) and AcroMetrix Oncology Hotspot Control (AcroMetrix, Thermo Fisher Scientific, Carlsbad, CA, USA). All mutations covered by the panel (four in total) were detected in the HDx control, except for one mutation in the *EGFR* gene with an allelic frequency (AF) that was under the limit of detection for the technique (0.9% as reported by the company). Analysis of the AcroMetrix reference DNA detected 123 variants (within the same range of AF reported by the company) of the 146 that could be potentially detected by our custom panel. The AcroMetrix synthetic DNA was designed and engineered to contain a high number of variants, some of which are linked in the same haplotype. As a result, the variants that are located in the primer sequence interfere with the amplification of the ones that are located in the insert of the amplicon. A detailed review of the 23 remaining variants that were not detected (or detected at lower frequencies) in this analysis revealed that all were within amplicon inserts that had a variant in the primer sequence. Therefore, these 23 variants are not expected to be amplified and were not included in the measures of sensitivity, as recently reported [19]. Additionally, we tested two available tumor samples with known mutations in exon 7 of *PTEN*, which are located in the same amplicons as two non-detected AcroMetrix variants. Both mutations were detected by the glioma panel at similar AFs as those obtained with the Ion AmpliSeq Cancer Hotspot Panel (CHP) (9% and 38% in the glioma panel versus 13% and 39% in the CHP, respectively) (Appendix A).

We then performed comparative analyses using the 22 glioma samples that were assessed for mutation detection using the CHP. The glioma panel detected all the mutations that were previously identified by the CHP (33/33, 100%). Remarkably, the glioma panel allowed us to detect seven samples (31.8%) with mutations in the *EGFR* gene that were not previously detected using the CHP, because this panel does not cover the whole gene. Furthermore, the AFs were highly correlated between both panels (*R* = 0.99), even for those variants with an AF close to the limit of detection (4%) (Appendix A). When all analyzed samples for the validation assay were taken into account, the sensitivity of the panel was 99.4%, with 159 of the total 160 mutations detected. Only one variant, with an AF under the limit of detection of the technique, was not detected.

We validated the specificity of the panel for glioma testing by using 12 samples from different tumors. We did not detect any mutation, except for four tumors that presented *PTEN*, *TP53*, or *BRAF* mutations previously tested by using the CHP.

Furthermore, we correlated the *IDH1* and *H3F3A* NGS mutation data with the results of the *IDH1* R132H and H3.3–K27M immunohistochemical staining. All *IDH1* p.R132H and *H3F3A* p.K27M mutations that were identified by using the NGS panel were validated by immunohistochemistry (Table 1).

### 2.3. Detection of the 1p/19q Loss of Heterozygosity Using the Glioma Next-Generation Sequencing Panel

To validate the 1p/19q codeletion test, we used 12 non-tumor samples and 65 tumor samples that were already tested by FISH. Representative 1p/19q results obtained from the analysis of the glioma panel are shown in Figure 1. No allelic imbalances were observed in the non-tumor samples (100% specificity), and concordant positive results were obtained for 21 of the 22 glioma samples (95.4% sensitivity) (Table 1). The only discordant case (#117) could be explained by the existence of two different cell populations, as described in the FISH report; alternatively, it may be a false-positive FISH result. However, this case was diagnosed as a glioblastoma (GBM) *IDH*-wild-type (*IDH* wt), and no mutations or CNVs were detected by the NGS glioma or CHP panels. A very good agreement was observed between both tests (Cohen Kappa statistic = 0.964).

Among the 31 gliomas that did not show a 1p/19q codeletion by FISH, 28 showed neither patterns of 1p/19q loss of heterozygosity (LOH) by NGS, as defined by our criteria (three were non-informative) (Appendix A). Similarly, the exclusive 19q-FISH loss was confirmed by LOH–NGS in five of six cases (one non-informative). However, different patterns were identified through the use of FISH and NGS regarding the six 1p36-FISH deleted cases, since the LOH of 1p was not identified in any of them. Three of these six tumors presented 1q-LOH, suggesting a gain of 1q that would explain the reported 1p-FISH loss.

It is important to note that a total of four samples were non-informative for the 1p/19q NGS test in the validation set, and that all of these samples showed *EGFR* amplification. In the whole sample set, 13 cases were noninformative for the 1p/19q analysis, all of which displayed *EGFR* amplification. In cases with high level copy number gains of *EGFR* gene, the amplicons that cover the gained region may be preferentially amplified, which can impede the assay’s ability to reach sufficient coverage of the other locations—for instance, the single nucleotide polymorphisms (SNPs) in chromosomes 1 and 19.

### 2.4. Detection of EGFR Amplification with the Glioma Next-Generation Sequencing Panel

High-levels of *EGFR* amplification were detected by comparing the read coverage of each sample with a baseline performed on 12 formalin-fixed, paraffin-embedded (FFPE), non-tumor samples. *EGFR* gene amplification was tested by chromogenic in situ hybridization (CISH) in 26 cases, and concordant results were obtained for all of them (16 amplified and 10 negative cases) (Table 1). Perfect agreement was observed between tests (Cohen Kappa statistic = 1).

### 2.5. Diagnosed Cases Using the 2016 Updated World Health Organization Classification with the Glioma Next-Generation Sequencing Panel

After validation of the panel, we applied it to a prospective series of 91 brain lesion samples. Given the potential difficulties of a histological diagnosis, we decided to apply the panel to gliomas and brain samples that raised difficulties during their differential diagnosis (such as one glioneuronal tumor, one subependymal giant-cell astrocytoma (SEGA), one undifferentiated tumor, and five cases with suspicion of brain tumor) (Appendix A). We classified all of the 143 samples that were analyzed with this NGS panel, according to the updated 2016 WHO classification system (Figure 2). Twenty-three patients were classified as *IDH*-mutant, 1p/19q-codeleted oligodendroglioma; 25 *IDH* mutant astrocytoma; and 55 *IDH* wt astrocytoma. Additionally, five cases presented with the *H3F3A* K27M mutation, and were therefore defined as diffuse, midline, H3 K27M-mutant glioma. Taking into account the prospective glioma series of cases, the groups obtained using the integrated molecular and histopathologic diagnostic showed frequencies not significantly different from those already described [5].

Among the 55 astrocytoma *IDH* wt tumors, all five *IDH* wt, WHO grade II tumors had mutations that altered telomere function, including four that had *TERT* promoter (p*TERT*) mutations, and one that had an *ATRX* truncating mutation. Nevertheless, while two of these tumors had accompanying *EGFR* or *PTEN* alterations, the only alteration identified across three of them was the telomere alteration (p*TERT* or *ATRX*). Regarding the high-grade *IDH* wt tumors, most of the cases (83%, 43/49) had alterations in either *EGFR*, *PTEN*, or p*TERT*. Three GBM cases presented with mutations in *TP53* as a driver event, and one exceptional pediatric case showed multiple gene alterations, suggesting Li–Fraumeni syndrome (case 95). We did not find any mutations in three tumors, although abnormal LOH patterns were identified in chromosomes 1 or 19 for two of them.

In total, we did not detect any mutations or CNV by using this NGS panel in 18 tumors (Figure 2): 5 samples subjected to the NGS study with suspicion of brain tumor, one ganglioglioma, one glioneuronal tumor, one SEGA, seven pilocytic astrocytomas, and the three previously mentioned GBMs. Further analyses of these samples by additional NGS panels revealed *KIAA1549*–*BRAF* fusion in six pilocytic astrocytomas and one *PTPN11* mutation in the glioneuronal tumor. No other alterations were identified in the 11 cases through the use of these additional panels.

## 3. Discussion

The recently updated WHO classification system for CNS tumors integrates histological features with molecular data for the diagnosis of gliomas, and involves testing numerous biomarkers, such as gene mutations, CNVs, and rearrangements. Gene-targeted NGS is an attractive approach for the clinical setting, as it allows for the simultaneous testing of multiple markers with high specificity, sensitivity, and a good cost-effectiveness ratio. Nevertheless, it is necessary to balance the size of these panels and the amount of information obtained by them, in order to address the increasing number of samples that need to be analyzed by NGS and to integrate these analyses into the routine practices of a pathology department. Here, we report the design and validation of an NGS panel for the diagnosis of gliomas that was successfully applied to routinely processed tissue samples in our laboratory. This panel enables the testing of gene mutations and CNVs that are necessary to establish a differential diagnosis of glioma. Furthermore, the size of this panel enables it to be integrated into routine clinical practice without any delayed turnaround time (TAT).

The amplicon-based technology employed here has a number of limitations, such as problems in the uniformity of coverage, amplification biases associated with GC content and length, or even false positives and false negatives [20]. This method, however, has been validated in different studies, and it is currently being used for routine molecular diagnosis of gliomas in other laboratories [16,18]. This method is less time-consuming, and allows for adequate TAT in a clinical setting in which poor quality and a low quantity of DNA from FFPE samples is available.

Importantly, the WHO recommends that detection of the 1p/19q codeletion includes whole-arm 1p and 19q testing [5]. On the one hand, the interstitial deletions that are frequent in astrocytomas do not have the same prognostic value as the 1p/19q whole-arm codeletion [21]. On the other hand, the proposed mechanism for the 1p and 19q loss in oligodendrogliomas, which involves the loss of the derivative chromosome carrying the 1p and 19q arms after a t(1;19) unbalanced translocation, involves the loss of the whole 1p and 19q arms [22,23]. The methodology reported here to test the 1p/19q codeletion was designed to test the whole-arm loss of chromosomes 1 and 19. In contrast to other panels used for the diagnosis of gliomas that are not suitable to detect this alteration [18], this panel reliably detects the 1p/19q codeletion. Furthermore, other glioma-specific panels do not fully cover chromosomes 1 and 19, which impedes their ability to distinguish between a whole chromosome loss and a whole arm loss, while others are not specifically designed to detect this alteration, or are so large that they are not suitable for use in the routine clinical practice of many pathology departments [15,16,17,24,25]. 

We report a good correlation between the detection of a 1p/19q loss by FISH and the presence of LOH, as determined by our NGS panel. We identified only one case with discordant 1p/19q codeletion results between the NGS and the FISH assays (2% out of a total of 60 cases). Regarding this case, the morphologic diagnosis of GBM and the absence of an *IDH* mutation may suggest that this is a false-positive FISH result, although the absence of any other alterations suggests the presence of other infrequent molecular alterations. A previous study that correlated FISH and whole-exome sequencing or array comparative genomic hybridization (aCGH) in GBMs showed that approximately 14% (11/80) of discordant cases were mainly due to false-positive FISH results, which are frequently caused by partial 1p or 19q losses [26]. The FISH technique uses loci-specific probes to detect the 1p/19q codeletion, and is therefore not suitable to detect whole chromosome arm deletions. Furthermore, the different criteria for scoring the 1p or 19q deletion (percentage of deleted cells versus the ratio test/control probe) may limit its detection by FISH [26]. Finally, another study showed that up to 20% (10/49) of the discordances between microsatellite-based LOH and SNP-based LOH detected by NGS are caused by a partial LOH of 19q that was not identified in the microsatellite analyses [24].

The NGS method used here to detect the 1p/19q codeletion does not have the FISH- and microsatellite-based LOH limitations, and is able to detect whole chromosome arm LOH through strong coverage of both chromosomes. However, in contrast to FISH, this NGS–LOH method is based on the detection of the LOH of SNPs that are located along chromosomes 1 and 19, and does not provide an absolute measure of the ploidy status of the tumor.

Furthermore, the panel designed here allows the distinction between losses of the entire chromosome, of the whole chromosome arm, or interstitial chromosomal losses. Notably, we identified 59 samples in the whole sample set (data not shown) that had partial or complete LOH along chromosomes 1 or 19, or both. By using microsatellite analyses, a previous study showed that the LOH on chromosome 19q was frequently found in secondary glioblastomas (54%), but was rarely detected in primary glioblastomas [27]. This alteration was also observed in other reports [26], but additional research is needed to determine its clinical relevance.

The gene-targeted NGS panel described here enables the successful identification of alterations for tumor diagnosis in accordance with the 2016 WHO classification. We received five cases for differential diagnosis where histology could not confirm the presence of a glial tumor, and the molecular findings suggested a lack of tumoral alterations (mutations and CNVs) for all these cases. One additional case was an undifferentiated metastatic tumor that presented with a *TP53* mutation. The absence of other alterations frequently found in gliomas suggests that it was not a glial primary tumor. Remarkably, three GBMs (approximately 2% of the cases) did not present with any of the genetic events that are frequently found in gliomas, and their diagnosis therefore relied entirely on the histological findings. Our results highlight that even if some additional markers, such as *pTERT* or *ATRX*, are included in the future editions of the WHO classification system, a small percentage of cases will still require additional molecular testing, and their diagnosis will still rely entirely on the morphological evaluation.

## 4. Materials and Methods

### 4.1. Samples

A retrospective collection of samples, which consisted of 52 glioma samples from 47 patients and 12 additional non-glioma cases, was performed for the validation of the glioma panel. These samples were selected according to the availability of NGS results obtained by the Ion AmpliSeq CHP (v2, Thermo Fisher Scientific, Carlsbad, CA, United States) and the availability of the 1p/19q codeletion status, as determined by FISH (both techniques are accredited in our laboratory, BELAC n°387-MED, NBN EN ISO 15189:2012). Control FFPE samples were comprised of 12 nontumor tissues and two tumor samples with known mutations. Two reference standard DNAs were also used as mutation controls: HDx (Horizon Discovery, Cambridge, United Kingdom) and AcroMetrix (Thermo Fischer). Prospective samples were analyzed between November 2017 and September 2018 (91 samples analyzed for the routine diagnosis of 82 patients). For tracking the turnaround time (TAT), we retrieved the status data from the laboratory information system. All samples were obtained from FFPE tissues. Sample and patient diagnosis information for the validation and prospective series are provided in Appendix A.

### 4.2. Next-Generation Sequencing Panel

We designed and constructed an NGS panel for glioma diagnosis that targets 14 glioma-related genes (hotspots, selected exons, or complete coding sequence) and 55 SNPs that are located along the full length of chromosomes 1 and 19 (Table 2). Target genes were selected based on the fact that they were frequently mutated in glioma tumors, and according to the WHO diagnostic requirements for molecular testing in gliomas [5]. These SNPs were selected based on the high polymorphic value of their minor allele frequency, as reported on the National Center for Biotechnology Information (NCBI) dbSNP database, as well as from previous reports [24,28]. The average distance between SNPs is 5.2 Mb in 1p, 8.0 Mb in 1q, 1.9 Mb in 19p, and 2.1 Mb in 19q. To design the panel, we used the Ion AmpliSeq Designer software (Thermo Fisher). Validation was performed using the previously described commercially available CHP that covers hotspot mutations for 50 genes.

DNA from an FFPE tumor and normal tissues were extracted using a QIAamp DNA FFPE Tissue Kit (Qiagen, Manchester, United Kingdom) or the Maxwell RSC DNA FFPE Kit (Promega Corporation, Madison, WI, United States) following the manufacturer’s instructions. The DNA (and library) yield was quantified using a Qubit 2.0 Fluorometer (Thermo Fisher) or the QuantusTM Fluorometer (Promega). Library construction was performed using an Ion AmpliSeq Library Kit v2.0 with 1 to 10 ng of input DNA, depending on the amount of available DNA. Libraries were multiplexed, submitted to emulsion PCR on the Ion Chef System (Thermo Fisher), and sequenced using an Ion Torrent Personal Genome Machine (Thermo Fisher).

### 4.3. Computational Analysis of Next-Generation Sequencing Data

The sequences were aligned to the hg19 reference genome, and variant calling was performed using the Torrent Mapping Alignment Program (Torrent Suite v.5.6, Thermo Fisher). The following quality metrics were taken into account for each sequenced sample: (1) total mapped reads (≥200,000) and (2) average base coverage depth (≥1000×, optimal; ≥500× and <1000×, suboptimal; <500×, low quality). For mutation detection, variants were filtered out basing on our previous report [29]: (1) fewer than 100 reads; (2) a mutant allele coverage of fewer than 30 reads; (3) a location outside of the exonic or splice regions; (4) synonymous variants; (5) variants present in the population with a minor allele frequency greater than 1%, according to the 1000 Genomes project (dbSNP build id 138); and (6) variants systematically present in different runs or in all the samples from the same run (sequencing artifacts). All the remaining variants were visualized using the Integrative Genomic Viewer (IGV; Broad Institute) [30], and were manually reviewed in COSMIC and ExAc databases as well as in the scientific literature to exclude polymorphisms or nonpathogenic variants.

To detect copy number variations (CNVs), we used the Ion reporter software (Thermo Fisher). The algorithm normalizes the read coverage across amplicons to predict copy numbers, by comparing the sample reads to a baseline coverage constructed from 12 male control diploid DNA samples. According to the manufacturer’s instructions, and considering the number of amplicons covering each gene, the CNVs in this panel using this method were only evaluated for *EGFR* gene amplification. CNV data were filtered to exclude regions with confidences lower than 10, as recommended by the manufacturer (Thermo Fisher).

To detect 1p/19q LOH, we firstly applied a quality criterion based on the SNP coverage. The test was considered optimal, suboptimal, or non-informative, according to the number of SNPs that were covered by fewer than 250 reads (Table 3).

Secondly, the allelic frequencies (AF) for each SNP (with more than 250×) were annotated. Homozygous SNPs with the same nucleotide as that of the reference genome will have an AF of approximately 100%, while homozygous SNPs with a nucleotide that differs from the reference genome will have an AF of approximately 0%. Heterozygous SNPs will have an AF of approximately 50%. However, because NGS provides a semi-quantitative measure based on the number of reads [19], we established the following confidence intervals: 90–100% or 0–10% for homozygous markers, and 40–60% for heterozygous markers. These confidence intervals were defined based on the analysis of 12 nontumor samples. Imbalances of 1p and 19q markers due to LOH were scored when their AFs were outside the established ranges for homozygosity or heterozygosity (i.e., 10–40% or 60–90%) (Figure 1).

We defined the criterion for the 1p/19q codeletion as the absence of any heterozygous markers in these chromosomal arms, together with the presence of at least one heterozygous marker in the opposite arm. No codeletion was scored if at least one heterozygous marker was present in 1p or 19q. A whole chromosome arm LOH is observed when there are no heterozygous markers present in either arm.

### 4.4. In Situ Hybridization Techniques

Fluorescence in situ hybridization of paraffin-embedded tissue sections was performed to detect chromosomal losses of 1p and 19q using the probes LSI 1p36/1q25 and 19p13/19q13 (Vysis Inc., Downers Grove, IL, United States), as previously described [9]. The codeletion was considered to be positive in cases where the ratio of red (1p or 19q) versus green (1q or 19p) signals were lower than 0.8 after scoring at least 50 nuclei, according to our BELAC ISO 15189 accredited laboratory protocol.

CISH was performed to detect *EGFR* amplification by using the Zytodot 2C SPEC EGFR/CEN7 probes (ZytoVision GmbH, Bremerhaven, Germany), according to the manufacturer’s recommendations. Amplification of *EGFR* was positively scored when the ratio of *EGFR* versus the centromeric signal was greater than 2 after counting at least 50 nuclei.

### 4.5. Immunohistochemistry

For validation purposes, immunohistochemical staining was used to detect *IDH1* R132H (mouse monoclonal antibody; dilution 1:100, DIA H09, CKPIIIGHHAYGD; Dianova GmbH, Hamburg, Germany) and H3.3-K27M (rabbit polyclonal; dilution 1:1000, ABE419; Merck-Millipore, Billerica, United States) using an automated immunostainer (Dako Omnis, Agilent Technologies, Santa Clara, CA, United States), according to the manufacturer’s recommendations.

## 5. Conclusions

In summary, we report an accurate and sensitive method to detect mutations and copy number alterations, including the 1p/19q codeletion and the *EGFR* amplification, using a rapid and cost-effective approach that enables the correct classification of gliomas and facilitates the diagnosis of difficult cases.

## Figures and Tables

**Figure 1 cancers-11-00773-f001:**
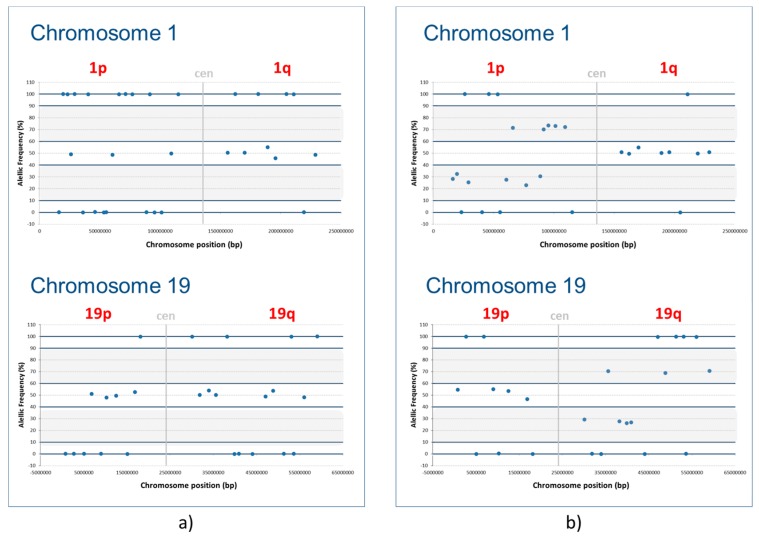
Analysis of the 1p/19q codeletion through the detection of single-nucleotide polymorphism (SNP)-based loss of heterozygosity (LOH), using the designed next-generation sequencing panel. (**a**) Representative 1p/19q non-codeleted sample. (**b**) Representative 1p/19q codeleted sample. For each sample and chromosome, the allelic frequency is represented according to the SNP location. The established ranges for homozygosity were defined between 0% and 5% (same allele as that of the reference genome) and between 90% and 100% (different allele than that of the reference genome). The range of heterozygosity was defined as being between 40% and 60% of allelic frequency (AF). The LOH for an SNP was defined as when the AF was outside of the established ranges for homozygosity or heterozygosity (gray areas). A sample was scored as having 1p/19q LOH when all 1p and 19q markers had LOH or were homozygous, and at least one marker in the opposite arm (1q and 19p) were heterozygous.

**Figure 2 cancers-11-00773-f002:**
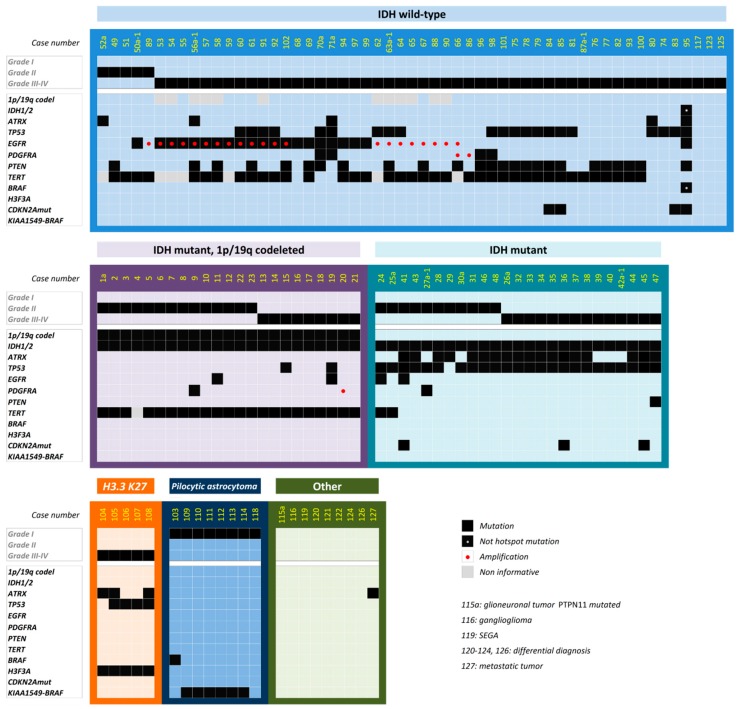
Genomic alterations identified in the whole series. Samples (columns) are grouped by 2016 World Health Organization (WHO) criteria. The WHO grade and somatic mutations are indicated by black squares.

**Table 1 cancers-11-00773-t001:** Summarized results of the next-generation sequencing (NGS) validation performed using orthogonal methods.

NGS Glioma Panel	Orthogonal Method	
	Fluorescence ISH ^1^	
	1p/19q codel	No codel
1p/19q LOH	21	0
No LOH	1	38
Non-informative	0	5
	Chromogenic ISH ^1^	
	*EGFR* amplification	No *EGFR* amplification
*EGFR* amplification	16	0
No *EGFR* amplification	0	10
	*IDH1* R132H IHC ^2^	
*IDH1/2* mutant	Positive	Negative
*IDH1* p.R132H	34	0
*IDH1* p.R132C/S/G	0	3
*IDH2* p.R172K	0	3
*IDH1/2* wild-type	0	48
	H3.3 K27M IHC ^2^	
	Positive	Negative
*H3F3A* mutant	3	0
*H3F3A* wild-type	0	9

^1^ ISH: in situ hybridization; ^2^ IHC: immunohistochemistry.

**Table 2 cancers-11-00773-t002:** Genes and regions tested in the glioma panel.

Chromosome	Gene	RefSeq No.	Hotspot (Aminoacid Position)/Coding Sequence (% Sequence Covered)
chr1	*H3F3A*	NM_002107	Hotspot (K28, G35)
chr2	*ACVR1*	NM_001105	Hotspot (R206, Q207, A233, R258, G328, G356)
chr2	*IDH1*	NM_005896	Hotspot (R132)
chr4	*PDGFRA*	NM_006206	Frequently mutated exons (5-12, 14-15, 18, 21-23)
chr5	*TERT*	NM_001193376	Hotspot (promoter region, C228T, C250T)
chr6	*HIST1H3B*	NM_003537	Hotspot (K28)
chr6	*HIST1H3C*	NM_003531	Hotspot (K28)
chr7	*EGFR*	NM_005228	Whole coding region (100.0%)
chr7	*BRAF*	NM_004333	Hotspot (V600)
chr9	*CDKN2A*	NM_000077	Whole coding region (85.4%)
chr10	*PTEN*	NM_000314	Whole coding region (99.3%)
chr15	*IDH2*	NM_001289910	Hotspot (R172)
chr17	*TP53*	NM_000546	Whole coding region (99.9%)
chrX	ATRX	NM_000489	Whole coding region (100.0%)
**Chromosome**	**Gene**	**Copy Number Variation**	
chr1	30 SNPs	1p/19q Codeletion	
chr19	25 SNPs		
chr7	*EGFR*	Amplification	

**Table 3 cancers-11-00773-t003:** Quality criteria applied to the single nucleotide polymorphism (SNP) analysis.

	Number of SNPs with Coverage of <250×		
	1p	19q	1q or 19p
Noninformative	≥11	≥8	≥6
Suboptimal	≤7	≤6	≤5
Optimal	<7	<6	<5
Total number of SNPs	20	15	10

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
