# Peer review of "Design and Validation of a Gene-Targeted, Next-Generation Sequencing Panel for Routine Diagnosis in Gliomas"

_cancers, 2019, doi:10.3390/cancers11060773_

Round 1

Reviewer 1 Report

D' Haene et al. designed a 14-gene next-generation sequencing panel which aims to specifically diagnose glioma. However, the authors did not really address the specificity of this panel is actually glioma specific. They compared the mutations of the tumors vs non-tumors but lack the analysis of glioma vs other type of cancers. This analysis/justification is needed for publication.

Minor comments:

Line 52-53:"require assessing the tumor for IDH1 and IDH2 (IDH), H3F3A and BRAF mutations, as well as the 1p/19q codeletion" citations are needed.

the authors mentioned "noninformative" throughout the paper, some of which are due to "low quality". Should these samples be discarded in the original data filtering?

Author Response

Reviewer 1

Firstly we would like to really thank the reviewer for her/his comments and suggestions on the manuscript that have helped us to improve the quality of our work.

Please, find in the PDF file attached our point by point responses. 

Reviewer 2 Report

The study presented in this manuscript aimed to validate a diagnostic tool for a specific type of cancer. The authors decided to design a gene panel that could be a reliable diagnostic tool not offered yet by commercial companies. The selection criteria is well applied in the design, according to the WHO (2016). However, the sequencing strategy selected forces other possible interested laboratories to use both Ampliseq and Ion Torrent platforms. False negative SNP detection is known as part of the troubleshooting in Amplicon-based strategies in FFPE samples, while Ion Torrent platform presents known problems in the homogeneity of the distribution of read

s among samples. My questions are then:

1- Why amplicon-based estrategy has been selected for the gene panel design? Which is the overlapping between amplicons? As seen in Appendinx A, Figure 1, since not a real overlapping fragment exists between both amplicons, some of the target bases are not covered at all, and for that, false negatives are prone to arise.

2- Have the authors analysed the data in an alternative way other that the Ion Reporter software? I would like to see the number of reads obtained per sample and the distribution of depth accross target regions in order to check the homogeneity in the distribution of information.

3- LOH is an indirect way of identifying losses in chromosomal regions. in addition, the authors stated that CNVs could only be checked for EGFR gene due to limitations in the analysis pipeline. Have the authors used other tools to check CNVs?

4- What is the average distance between SNPs across chr1 and chr19?

5- The authors stated that 16 cases were noninformative for the 1p/19q analysis: 4 of them for low quality and the others due to the EGFR amplification. Can the author explain it in more detail? Have the author used alternative analysis strategies? Have these samples showed a different pattern of quality during the library preparation steps? Maybe the author can add more appendix information related to sequencing stats.

6- The author stated that one variant was missed due to AF = 0.9%. Which was the depth obtained in this region? Was the number of reads bearing the variant allele greater of 10? Maybe the authors should applied a different criteria set during the filtering of variants.

Author Response

Reviewer 2

Firstly we would like to really thank the reviewer for her/his comments and suggestions on the manuscript that have helped us to improve the quality of our work.

Please, find in the PDF file attached our point by point responses. 

Reviewer 3 Report

- In this work, D’Haeneet al.  described the validation of a custom NGS panel for glioma diagnosis and the first prospective study performed in this pathology by NGS. On my opinion, the manuscript is well written and easy to understand. I only have minor comments.

- Define acronyms upon the first usage in the text. The results section is above than the material and methods section.

- Figure and tables from supplementary information are not well listed. [Figure 1 (Figure S1) and tables (Table S1, table S2)].

- Page 4, line 152. “We classified all the samples that were analyzed with this NGS panel according to the updated 2016 WHO classification system”. Within the text the authors described that 91 samples were prospectively analyzed but classification was done in a larger cohort of patients. Please correct the numbers. 

“Twenty-three patients were oligodendroglioma, IDH-mutant and 1p/19q-codeleted; 25 were astrocytoma,IDH mutant; and 55 were astrocytoma IDH wt. Additionally, five cases presented with 1 the H3F3A K27M mutation and were therefore defined as diffuse midline glioma, H3 K27M-mutant”.

Are these frequencies found normal? Please add frequencies and compared the data with data already published.

- Page 6, line 195. Could you add this information also in the results section? “Furthermore, the size of this panel enables it to be integrated into routine clinical practice without any delayed turnaround time”.

- Page 7, line 259. Please specify the laboratory with ISO accreditation.

- Table 1. Please add Cohen Kappa test: Correlation between orthogonal testing and NGS results.

Author Response

Reviewer 3

Firstly we would like to really thank the reviewer for her/his comments and suggestions on the manuscript that have helped us to improve the quality of our work.

Please, find in the PDF file attached our point by point responses. 

Round 2

Reviewer 1 Report

the authors have addressed my previous comments/concerns and i recommend this paper to be accepted for publication.

Reviewer 2 Report

Thanks to the authors for taking into consideration my comments and suggestions.